# Bounded Myopic Adversaries for Deep Reinforcement Learning Agents

## Abstract

Adversarial attacks against deep neural networks have been widely studied. Adversarial examples for deep reinforcement learning (DeepRL) have significant security implications, due to the deployment of these algorithms in many application domains. In this work we formalize an optimal myopic adversary for deep reinforcement learning agents. Our adversary attempts to find a bounded perturbation of the state which minimizes the value of the action taken by the agent. We show with experiments in various games in the Atari environment that our attack formulation achieves significantly larger impact as compared to the current state-of-the-art. Furthermore, this enables us to lower the bounds by several orders of magnitude on the perturbation needed to efficiently achieve significant impacts on DeepRL agents.

## 1 Introduction

Deep Neural Networks (DNN) have become a powerful tool and currently DNNs are widely used in speech recognition (Hannun et al., 2014), computer vision (Krizhevsky et al., 2012), natural language processing (Sutskever et al., 2014), and self learning systems as deep reinforcement learning agents (Mnih et al. (2015), Mnih et al. (2016), Schulman et al. (2015), Lillicrap et al. (2015)).

Along with the overwhelming success of DNNs in various domains there has also been a line of research investigating their weaknesses. Szegedy et al. (2014) observed that adding imperceptible perturbations to images can lead a DNN to misclassify the input image. The authors argue that the existence of these so called adversarial examples is a form of overfitting. In particular, they hypothesize that a very complicated neural network behaves well on the training set, but nonetheless, performs poorly on the testing set enabling exploitation by the attacker. However, they discovered different DNN models were misclassifying the same adversarial examples and assigning them the same class instead of making random mistakes. This led Goodfellow et al. (2015) to propose that the DNN models were actually learning approximately linear functions resulting in underfitting the data.

Recent work by Mnih et al. (2015) introduced the use of DNNs as function approximators in reinforcement learning, improving the state of the art in this area. Because these deep reinforcement learning agents utilize DNNs, they are also susceptible to this type of adversarial examples. Currently, deep reinforcement learning has been applied to many areas such as network system control (Jay et al. (2019), Chu et al. (2020), Chinchali et al. (2018)), financial trading Noonan (2017), blockchain protocol security Hou et al. (2019), grid operation and security (Duan et al. (2019), Huang et al. (2019)), cloud computing Chen et al. (2018), robotics (Gu et al. (2017), Kalashnikov et al. (2018)), autonomous driving Dosovitsky et al. (2017), and medical treatment and diagnosis (Tseng et al. (2017), Popova et al. (2018), Thananjeyan et al. (2017), Daochang & Jiang (2018), Ghesu et al. (2017)). A more particular scenario where adversarial perturbations might be of significant interest is a financial trading market where the DeepRL agent is trained on observations consisting of the order book. In such a setting it is possible to compromise the whole trading system with an extremely small subset of adversaries. In particular, the $\ell_1$-norm bounded perturbations dicussed in our paper have sparse solutions, and thus can be used as a basis for an attack in such a scenario. Moreover, the magnitude of the $\ell_1$-norm bounded perturbations produced by our attack is orders of magnitude smaller than previous approaches, and thus our proposed perturbations result in a stealth attack more likely to evade automatic anomaly detection schemes.

Considering the wide spectrum of deep reinforcement learning algorithm deployment it is crucial to investigate the resilience of these algorithms before they are used in real world application domains. Moreover, adversarial formulations are a first step to understand these algorithms and build generalizable, reliable and robust deep reinforcement learning agents. Therefore, in this paper we study adversarial attack formulations for deep reinforcement learning agents and make the following contributions:

- We define the optimal myopic adversary, whose aim is to minimize the value of the action taken by the agent in each state, and formulate the optimization problem that this adversary seeks to solve.

- We introduce a differentiable approximation for the optimal myopic adversarial formulation.

- We compare the impact results of our attack formulation to previous formulations in different games in the Atari environment.

- We show that the new formulation finds a better direction for the adversarial perturbation and increases the attack impact for bounded perturbations. (Conversely, our formulation decreases the magnitude of the pertubation required to efficiently achieve a significant impact.)

## 2 RELATED WORK AND BACKGROUND

### 2.1 ADVERSARIAL REINFORCEMENT LEARNING

Adversarial reinforcement learning is an active line of research directed towards discovering the weaknesses of deep reinforcement learning algorithms. Gleave et al. (2020) model the interaction between the agent and the adversary as a two player Markov game and solve the reinforcement learning problem for the adversary via Proximal Policy Optimization introduced by Schulman et al. (2017). They fix the victim agent's policy and only allow the adversary to take natural actions to disrupt the agent instead of using $\ell_p$-norm bound pixel perturbations. Pinto et al. (2017) model the adversary and the victim as a two player zero-sum discounted Markov game and train the victim in the presence of the adversary to make the victim more robust. Mandlekar et al. (2017) use a gradient based perturbation to make the agent more robust as compared to random perturbations. Huang et al. (2017) and Kos & Song (2017) use the fast gradient sign method (FGSM) to show deep reinforcement learning agents are vulnarable to adversarial perturbations. Pattanaik et al. (2018) use a gradient based formulation to increase the robustness of deep reinforcement learning agents.

### 2.2 ADVERSARIAL ATTACK METHODS

Goodfellow et al. (2015) introduced the fast gradient method (FGM)

$$x^* = x + \epsilon \cdot \frac{\nabla_x J(x, y)}{||\nabla_x J(x, y)||_p}, \tag{1}$$

for crafting adversarial examples for image classification by taking the gradient of the cost function $J(x, y)$ used to train the neural network in the direction of the input. Here $x$ is the input and $y$ is the output label for image classification. As mentioned in the previous section FGM was first adapted to the deep reinforcement learning setting by Huang et al. (2017). Subsequently, Pattanaik et al. (2018) introduced a variant of FGM, in which a few random samples are taken in the gradient direction, and the best is chosen. However, the main difference between the approach of Huang et al. (2017) and Pattanaik et al. (2018) is in the choice of the cost function $J$ used to determine the gradient direction. In the next section we will outline the different cost functions used in these two different formulations.

### 2.3 ADVERSARIAL ATTACK FORMULATIONS

In a bounded attack formulation for deep reinforcement learning, the aim is to try to find a perturbed state $s_{\text{adv}}$ in a ball

$$D_{\epsilon,p}(s) = \{s_{\mathrm{adv}} \mid \|s_{\mathrm{adv}} - s\|_p \leq \epsilon\}, \tag{2}$$

that minimizes the expected cumulative reward of the agent. It is important to note that the agent will always try to take the best action depending only on its perception of the state, and independent from the unperturbed state. Therefore, in the perturbed state the agent will still choose the action, $a^*(s_{\mathrm{adv}}) = \arg\max_a Q(s_{\mathrm{adv}}, a)$, which maximizes the state action value function in state $s_{\mathrm{adv}}$.

It has been an active line of research to find the right direction for the adversarial perturbation in the deep reinforcement learning domain. The first attack of this form was formulated by Huang et al. (2017) and concurrently by Kos & Song (2017) by trying to minimize the probability of the best possible action in the given state,

$$a^*(s) = \arg\max_a Q(s, a)$$
$$\min_{s_{\mathrm{adv}} \in D_{\epsilon,p}(s)} \pi(s_{\mathrm{adv}}, a^*(s)). \tag{3}$$

Note that $\pi(s, a)$ is the softmax policy of the agent given by

$$\pi_{\mathrm{T}}(s, a) = \frac{e^{\frac{Q(s, a(s))}{T}}}{\sum_{a_k} e^{\frac{Q(s, a_k)}{T}}}, \tag{4}$$

where T is called the temperature constant. When the temperature constant is not relevant to the discussion we will drop the subscript and use the notation $\pi(s, a)$. It is important to note that $\pi(s, a)$ is not the actual policy used by the agent. Indeed the DRL agent deterministically chooses the action $a^*$ maximizing $Q(s, a)$ (or equivalently $\pi(s, a)$). The softmax operation is only introduced in order to calculate the adversarial perturbation direction. Lin et al. (2017) formulated another attack with unbounded perturbation based on the Carlini & Wagner (2017) attack where the goal is to minimize the perturbation subject to choosing any action other than the best.

$$\min_{s_{\mathrm{adv}} \in D_{\epsilon,p}(s)} \quad \|s_{\mathrm{adv}} - s\|_p$$
$$\text{subject to} \quad a^*(s) \neq a^*(s_{\mathrm{adv}}).$$

Pattanaik et al. (2018) formulated yet another attack which aims to maximize the probability of the worst possible action in the given state,

$$a_w(s) = \arg\min_a Q(s, a)$$
$$\max_{s_{\mathrm{adv}} \in D_{\epsilon,p}(s)} \pi(s_{\mathrm{adv}}, a_w(s)), \tag{5}$$

and further showed that their attack formulation (5) is more effective than (3).

Pattanaik et al. (2018) also introduce the notion of targeted attacks to the reinforcement learning domain. In their paper they take the cross entropy loss between the optimal policy in the given state and their adversarial probability distribution and try to increase the probability of $a_w$. However, just trying to increase the probability of $a_w$ in the softmax policy i.e. $\pi(s_{\mathrm{adv}}, a_w)$ is not sufficient to target $a_w$ in the actual policy followed by the agent. In fact the agent can end up in a state where,

$$\pi(s_{\mathrm{adv}}, a_w) > \pi(s, a_w)$$
$$a_w \neq \arg\max_a \pi(s_{\mathrm{adv}}, a). \tag{6}$$

Although $\pi(s_{\mathrm{adv}}, a_w)$ has increased, the action $a_w$ will not be taken. However, it might be still possible to find a perturbed state $s'_{\mathrm{adv}}$ for which,

$$\pi(s_{\text{adv}}, a_w) > \pi(s'_{\text{adv}}, a_w)$$
$$a_w = \arg\max_a \pi(s'_{\text{adv}}, a). \tag{7}$$

Therefore, maximizing the probability of taking the worst possible action in the given state is not actually the right formulation to find the correct direction for adversarial perturbation.

## 3   OPTIMAL MYOPIC ADVERSARIAL FORMULATION

To address the problem described in Section 2.3 we define an *optimal myopic adversary* to be an adversary which aims to minimize the value of the action taken by the agent myopically for each state. The value of the action chosen by the agent in the unperturbed state is $Q(s, a^*(s))$, and the value (measured in the unperturbed state) of the action chosen by the agent under the influence of the adversarial observation is $Q(s, a^*(s_{\text{adv}}))$. The difference between these is the actual impact of the attack. Therefore, in each state $s$ the optimal myopic adversary must solve the following optimization problem

$$\arg\max_{s_{\text{adv}} \in D_{\epsilon,p}(s)} [Q(s, a^*(s)) - Q(s, a^*(s_{\text{adv}}))]. \tag{8}$$

By (4), we may rewrite (8) in terms of the softmax policies $\pi(s, a)$,

$$\arg\max_{s_{\text{adv}} \in D_{\epsilon,p}(s)} [\pi(s, a^*(s)) - \pi(s, a^*(s_{\text{adv}}))]. \tag{9}$$

Since $\pi(s, a^*(s))$ does not depend on $s_{\text{adv}}$, (9) is equivalent to solving

$$\min_{s_{\text{adv}} \in D_{\epsilon,p}(s)} \pi(s, \arg\max_a \{\pi(s_{\text{adv}}, a)\}). \tag{10}$$

### 3.1   EFFICIENT APPROXIMATION OF OPTIMAL MYOPIC ADVERSARIAL FORMULATION

Having an $\arg\max$ operator in the cost function is unpleasant, since it is non-differentiable. Instead we can approximate $\arg\max$ by decreasing the temperature T for $\pi_{\text{T}}(s_{\text{adv}}, a)$,

$$\lim_{\text{T}_{\text{adv}} \to 0} \pi_{\text{T}_{\text{adv}}}(s_{\text{adv}}, a) = \mathbb{1}_{\arg\max_{a'} \{\pi(s_{\text{adv}}, a')\}}(a). \tag{11}$$

The intuition for (11) is that, by (3), as we decrease the temperature the value of $e^{(Q(s,a^*)/\text{T}_{\text{adv}})}$ for the action $a^*$ which maximizes $Q(s, a)$ will dominate the other actions. Thus $\pi(s, \arg\max_a \{\pi(s_{\text{adv}}, a)\})$ can be approximated by,

$$\pi(s, \arg\max_a \{\pi(s_{\text{adv}}, a)\}) = \sum_a \pi(s, a) \cdot \mathbb{1}_{\arg\max_{a'} \{\pi(s_{\text{adv}}, a')\}}(a) \tag{12}$$

$$= \lim_{\text{T}_{\text{adv}} \to 0} \sum_a \pi(s, a) \cdot \pi_{\text{T}_{\text{adv}}}(s_{\text{adv}}, a) \tag{13}$$

Therefore, our original optimization problem can be expressed as,

$$\min_{s_{\text{adv}} \in D_{\epsilon,p}(s)} \pi(s, \arg\max_a \{\pi(s_{\text{adv}}, a)\}) = \min_{s_{\text{adv}} \in D_{\epsilon,p}(s)} \lim_{\text{T}_{\text{adv}} \to 0} \sum_a \pi(s, a) \cdot \pi_{\text{T}_{\text{adv}}}(s_{\text{adv}}, a). \tag{14}$$

In practice we will not be decreasing $\text{T}_{\text{adv}}$ to 0 as this would be equal to applying the non-differentiable argmax operation. Instead we will replace the $\arg\max$ operation with the approximation $\pi_{\text{T}_{\text{adv}}}(s, a)$

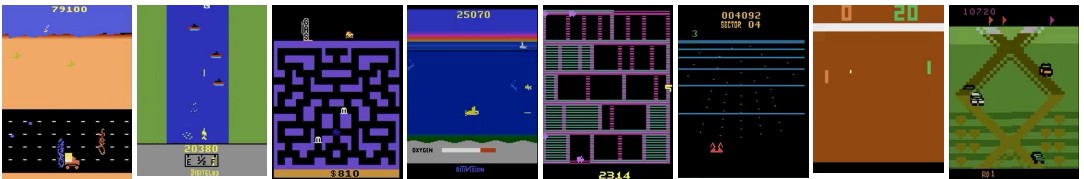

Figure 1: Games used in the experiments from Atari Arcade Environment. Games from left to right in order: Roadrunner, Riverraid, Bankkeist, Seaquest, Amidar, Beamrider, Pong and UpandDown.

for a small value of $T_{adv}$. In general it is not guarenteed that the minimum of the cost function in (14) with this approximation will be close to the minimum of the original cost function with the $\arg\max$ operation. To see why, note that this approximation is equivalent to first switching the limit and minimum in (14) to obtain,

$$\lim_{T_{adv} \to 0} \min_{s_{adv} \in D_{\epsilon,p}(s)} \sum_a \pi(s,a) \cdot \pi_{T_{adv}}(s_{adv}, a),$$ (15)

and second replacing the limit with a small value of $T_{adv}$. There are two possible issues with this approach. First of all, due to the non-convexity of the minimand, exchanging the limit and minimum may not yield an equality between (14) and (15). Secondly, even if this exchange is legitimate, it is not clear how to choose a sufficiently small value for $T_{adv}$ in order to obtain a good approximation to (15). However, we show in our experiments that using this approximation in the cost function gives state-of-the-art results.

There is another caveat which applies to all myopic adversaries including those in prior work. The Q-value is a good estimate of the discounted future rewards of the agent, only assuming that the agent continues to take the action maximizing the Q-value in future states. Since myopic attacks are applied in each state, this may make minimizing the Q-value a non-optimal attack strategy when the future is taken into account. There is also always the risk that the agent's Q-value is miscalibrated. However, our experiments show that our myopic formulation performs well despite these potential limitations.

## 3.2 EXPERIMENTAL SETUP

In our experiments we averaged over 10 episodes for each Atari game (Bellemare et al., 2013) in Figure 1 from the Open AI gym environment (Brockman et al., 2016). Agents are trained with Double DQN (Wang et al., 2016). We compared the attack impact of our Myopic formulation with the previous formulations of Huang et al. (2017) and Pattanaik et al. (2018). The attack impact has been normalized by comparing an unattacked agent, which chooses the action corresponding to the maximum state-action value in each state, with an agent that chooses the action corresponding to the minimum state-action value in each state. Formally, let $R_{max}$ be the average return for the agent who always chooses the best action in a given state, let $R_{min}$ be the average return for the agent who always chooses the worst possible action in a given state, and let $R_a$ be the average return of the agent under attack. We define the impact I, as

$$I = \frac{R_{max} - R_a}{R_{max} - R_{min}}.$$ (16)

This normalization was chosen because we observed that in Atari environments agents can still collect stochastic rewards even when choosing the worst possible action in each state until the game ends. See more details of the setup in Appendix A.2.

## 3.3 ADVERSARIAL TEMPERATURE

Based on the discussion in Section 3 we expect that as $T_{adv}$ decreases the function in (14) becomes a better approximation of the $\arg\max$ function. Thus this results in a higher attack impact as $T_{adv}$ decreases. However, after a certain threshold the function in (14) becomes too close to the $\arg\max$

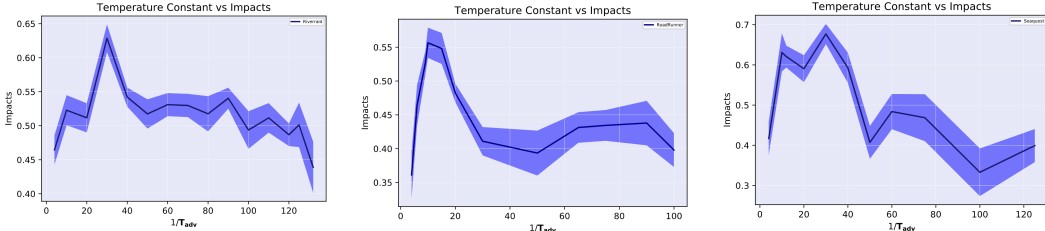

Figure 2: Attack impacts vs temperature constant for different games $\ell_2$-norm bound with $\epsilon = 10^{-8}$. Left: Riverraid. Middle: Roadrunner. Right: Seaquest.

Table 1: Left: Attack impacts for three attack formulations with $\ell_2$ norm bound and $\epsilon = 10^{-8}$. Right: Attack impacts for three attack formulations with $\ell_1$ norm bound and $\epsilon = 10^{-8}$.

| Games | Huang | Pattanaik | Myopic | Games | Huang | Pattanaik | Myopic |
|---|---|---|---|---|---|---|---|
| Amidar | 0.048±0.19 | 0.514±0.29 | 0.932±0.02 | Amidar | 0.039±0.22 | 0.502±0.32 | 0.941±0.03 |
| Bankheist | 0.111±0.14 | 0.373±0.15 | 0.624±0.08 | Bankheist | -0.014±0.07 | 0.0411±0.12 | 0.596±0.06 |
| Beamrider | 0.083±0.3 | 0.455±0.27 | 0.663±0.13 | Beamrider | -0.052±0.55 | 0.374±0.13 | 0.581±0.15 |
| Riverraid | -0.079±0.19 | 0.379±0.27 | 0.589±0.12 | Riverraid | 0.062±0.29 | 0.369±0.21 | 0.549±0.1 |
| RoadRunner | 0.187±0.16 | 0.387±0.16 | 0.557±0.11 | RoadRunner | 0.153±0.12 | 0.453±0.15 | 0.482±0.07 |
| Pong | 0.0±0.016 | 0.067±0.06 | 0.920±0.08 | Seaquest | 0.386±0.25 | 0.468±0.23 | 0.576±0.17 |
| Seaquest | 0.305±0.26 | 0.524±0.23 | 0.697±0.16 | Pong | 0.000±0.02 | 0.043±0.03 | 0.916±0.09 |
| UpNDown | 0.074±0.3 | 0.476±0.16 | 0.865±0.06 | UpNDown | -0.007±0.28 | 0.436±0.28 | 0.892±0.07 |

Figure 3: Attack impact vs logarithm base 10 of $\ell_2$-norm bound. Left: Amidar. Middle: Pong. Right: UpNDown.

function which is non-differentiable. Therefore, in practice, beyond this threshold the quality of the solutions given by the gradient based optimization will decrease, and so the attack impact will be lower. Indeed this can be observed in Figure 2. In our experiments we chose $T_{adv}$ to maximize the impact by grid search.

### 3.4 Impact Comparison for $\ell_p$-norm Bounded Perturbations

Table 1 shows the mean and standard deviation of the impact values under $\ell_1$ and $\ell_2$-norm bounds. The tables show that the proposed attack results in higher mean impact for all games and under all norms, and it results in a lower standard deviation almost always. In particular, this is an indication that our myopic attack formulation achieves higher impact more consistently than the previous formulations. More results on $\ell_\infty$-norm bound can be found in Section A.1 of the Appendix.

Figure 3 shows the attack impact as a function of the perturbation bound for each formulation in three games. As $\epsilon$ decreases, our myopic formulation exhibits higher impact relative to the other formulations. Recall that Goodfellow et al. (2015) argue that small adversarial perturbations shift the input across approximately linear decision boundaries learned by neural networks. Therefore, having a higher impact with smaller norm bound is evidence that our myopic formulation finds a better direction (i.e. one that points more directly at such a decision boundary) to search for the perturbation.

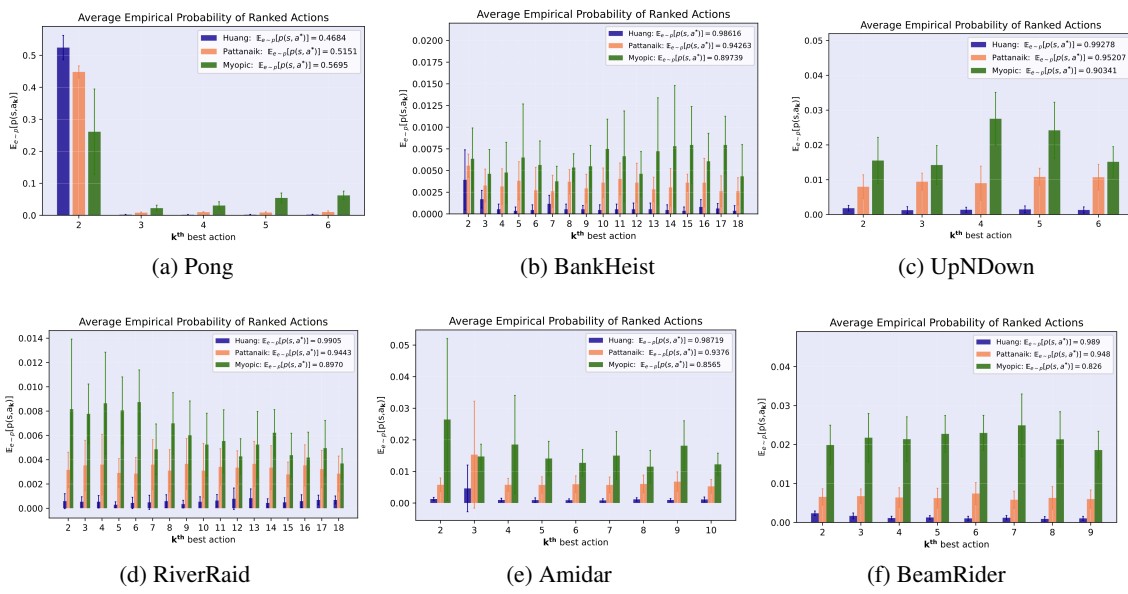

Figure 4: Up: Average empirical probabilities of ranked actions for three different formulations. Left: Bankheist. Middle: Pong. Right: UpNDown. Down: Expected probability of ranked actions of three different formulations. Left: Riverraid. Middle: Amidar. Right: BeamRider.

## 3.5 DISTRIBUTION ON ACTIONS TAKEN

In order to understand the superior performance of the proposed adversarial formulation it is worthwhile to analyze the distribution of the actions taken by the agents under attack. For each formulation, we recorded the empirical probability $p(s, a_k)$ that the attacked agent chooses the $k^{\text{th}}$ ranked action $a_k$. The ranking of the actions is according to their value in the unperturbed state. Without perturbation agents would always take their first ranked action, any deviation from this is a consequence of the adversarial perturbations. We average these values over 10 episodes to obtain the average empirical probability that the $k^{\text{th}}$ action is taken $\mathbb{E}_{e \sim \rho}[p(s, a_k)]$. Here $\rho$ is the distribution on the episodes $e$ of the game induced by the stochastic nature of the Atari games. We plot the results in Figure 4.

The legends in Figure 4 show the average empirical probabilities of taking the best action $a^*$. It can be seen that in general $\mathbb{E}_{e \sim \rho}[p(s, a^*)]$ is lower and $\mathbb{E}_{e \sim \rho}[p(s, a_w)]$ is higher for our Myopic formulation when compared to other attack formulations. We realized that the formulations which have higher impact at the end of the game might still be choosing the best action more often than the ones which have lower attack impact as shown in the legend of Figure 4 for the Pong game. In this case both Huang et al. (2017) and Pattanaik et al. (2018) cause the agent to choose the best action $a^*$ less frequently than our formulation, but still end up with lower impact. If we look at Figure 4 it can be seen that $\mathbb{E}_{e \sim \rho}[p(s, a^{2^{\text{nd}}})]$ is much higher for both Huang et al. (2017) and Pattanaik et al. (2018) compared to our Myopic formulation. However, the key to achieving a greater impact is to cause the agent to choose the lowest ranked actions more frequently. As can be seen from Figure 4, our Myopic formulation does this more successfully. More detailed results can be found in Appendix A.5.

## 3.6 STATE-ACTION VALUES OVER TIME

In this section we investigate state-action values of the agents over time without an attack, under attack with the Pattanaik et al. (2018) formulation, and under attack with our myopic formulation.

In Table 2 it is interesting to observe that under attack the mean of the state-action values over the episodes might be higher despite the average return being lower. One might think that an attack with greater impact might visit lower valued states on average. However, in our experiments we have found that the gap between $\mathbb{E}_{e \sim \rho}[Q(s, a^*(s))]$ and $\mathbb{E}_{e \sim \rho}[Q(s, a^*(s_{\text{adv}})]$ relative to the gap

Table 2: Average $Q$-values for the best, worst, the adversarial actions and impacts, loss in the $Q$-values caused by the adversarial influence, and impacts over the episodes.

| | $\mathbb{E}[Q(s, a^*(s))]$ | $\mathbb{E}[Q(s, a^*(s_{\text{ADV}}))]$ | $\mathbb{E}[Q(s, a_w)]$ | $Q_{\text{LOSS}}$ | IMPACT |
|---|---|---|---|---|---|
| RIVERRAID | | | | | |
| UNATTACKED | 9.724 | - | 8.911 | 0 | 0 |
| HUANG ET. AL | 9.656 | 9.653 | 8.8577 | 0.0039 | -0.079 |
| PATTANAIK ET. AL. | 10.092 | 10.072 | 9.341 | 0.0258 | 0.379 |
| MYOPIC | 10.013 | 9.977 | 9.283 | 0.0495 | 0.589 |
| SEAQUEST | | | | | |
| UNATTACKED | 9.724 | - | 8.911 | 0 | 0 |
| HUANG ET. AL | 10.6787 | 10.6787 | 10.100 | 0.0043 | 0.305 |
| PATTANAIK ET. AL. | 10.092 | 10.072 | 9.341 | 0.0267 | 0.524 |
| MYOPIC | 10.013 | 9.977 | 9.283 | 0.0506 | 0.697 |
| BANKHEIST | | | | | |
| UNATTACKED | 6.6863 | - | 5.9273 | 0 | 0 |
| HUANG ET. AL | 6.602 | 6.5998 | 5.907 | 0.0039 | 0.111 |
| PATTANAIK ET. AL. | 6.0517 | 6.0403 | 5.604 | 0.025 | 0.373 |
| MYOPIC | 5.9351 | 5.9153 | 5.5520 | 0.517 | 0.624 |
| AMIDAR | | | | | |
| UNATTACKED | 0.1188 | - | -0.1487 | 0 | 0 |
| HUANG ET. AL | 0.115 | 0.114 | -0.147 | 0.0049 | 0.048 |
| PATTANAIK ET. AL. | 0.139 | 0.138 | -0.163 | 0.0466 | 0.514 |
| MYOPIC | 0.2239 | 0.1953 | -0.2564 | 0.059 | 0.932 |

between $\mathbb{E}_{e \sim \rho}[Q(s, a^*(s))]$ and $\mathbb{E}_{e \sim \rho}[Q(s, a_w)]$ is a much more significant factor in determining the magnitude of the impact. Therefore, we define the quantity

$$Q_{\text{loss}} = \frac{\mathbb{E}_{e \sim \rho}[Q(s, a^*(s))] - \mathbb{E}_{e \sim \rho}[Q(s, a^*(s_{\text{adv}}))]}{\mathbb{E}_{e \sim \rho}[Q(s, a^*(s))] - \mathbb{E}_{e \sim \rho}[Q(s, a_w)]}. \tag{17}$$

Observe in Table 2 the magnitudes of $\mathbb{E}_{e \sim \rho}[Q(s, a^*(s))]$, $\mathbb{E}_{e \sim \rho}[Q(s, a_w)]$, and $\mathbb{E}_{e \sim \rho}[Q(s, a^*(s_{\text{adv}}))]$ are not strictly correlated to the impacts for each formulation. However, $Q_{\text{loss}}$ is always higher for our Myopic attack compared to the previous formulations. Recall that in Equation (3), our original optimization problem was designed to maximize the gap between $Q(s, a^*(s))$ and $Q(s, a^*(s_{\text{adv}}))$ in each state. The results in Table 2 are evidence both that we achieve this original goal and that solving our initial optimization problem for each state leads to lower average return at the end of the game. More results on this matter can be found in Appendix A.4.

The mean Q-value per episode under attack increases in some games. We believe that this may be due to the fact that the Q-value in a given state is only an accurate representation of the expected rewards of an unattacked agent. When the agent is under attack, there might be states with high Q-value which are dangerous for the attacked agent (e.g. states where the attack causes the agent to immediately lose the game). More results can be found in Appendix A.3.

## 4 FUTURE EXTENSIONS TO CONTINUOUS ACTION SETS

We note in this section that, at least from a mathematical point of view, it is possible to extend our formulation to continuous control tasks. Observe that the adversarial objective in (8) applies to continuous control problems too. The difficulty we are describing in (10) is not an issue in continuous control tasks. For instance, in Deep Deterministic Policy Gradient (DDPG) or Proximal Policy Optimization (PPO) the adversarial policy is already approximated by the actor network $\mu_\theta(s)$. That is: $a^*(s_{\text{adv}}) = \mu_\theta(s_{\text{adv}})$. Unlike in the case of discrete action sets, this means that the solution to the problem in (8) can be approximated through gradient descent following,

$$\nabla(Q(s, \mu_\theta(s_{\text{adv}}))) = \frac{\partial Q(s, a)}{\partial a}|_{a = \mu_\theta(s_{\text{adv}})} \cdot \nabla(\mu_\theta(s_{\text{adv}})), \tag{18}$$

where the gradient is taken with respect to $s_{\text{adv}}$. In a way it is actually easier to construct the adversarial examples in continuous control tasks because the learning algorithm already produces a differentiable approximation $\mu_\theta(s)$ to the argmax operation used in action selection. In our paper we focused on the derivation for the case of a discrete action set because the optimization problem in (10) is harder to solve.

## 5 CONCLUSION

In this paper we studied formulations of adversarial attacks on deep reinforcement learning agents and defined the optimal myopic adversary which incorporates the action taken in the adversarial state into the cost function. By introducing a differentiable approximation to the value of the action taken by the agent under the influence of the adversary we find a direction for adversarial perturbation which more effectively decreases the $Q$-value of the agent in the given state. In our experiments we demonstrated the efficacy of our formulation as compared to the previous formulations in the Atari environment for various games. Adversarial formulations are inital steps towards building resilient and reliable DeepRL agents, and we believe our adversarial formulation can help to set a new baseline towards the robustification of DeepRL algorithms.

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

## A   APPENDIX

### A.1   $\ell_\infty$-NORM BOUND

Table 3: Attack impacts for three attack formulations with $\ell_\infty$ norm bound and $\epsilon = 10^{-8}$.

| GAMES | HUANG | PATTANAIK | MYOPIC |
|---|---|---|---|
| AMIDAR | 0.208±0.14 | 0.290±0.37 | 0.536±0.25 |
| BANKHEIST | 0.197±0.12 | 0.410±0.09 | 0.527±0.16 |
| BEAMRIDER | 0.258±0.27 | 0.279±0.26 | 0.327±0.19 |
| RIVERRAID | -0.081±0.26 | 0.323±0.27 | 0.429±0.29 |
| ROADRUNNER | 0.125±0.16 | 0.445±0.13 | 0.553±0.11 |
| PONG | -0.002±0.01 | 0.065±0.03 | 0.086±0.04 |
| SEAQUEST | 0.344±0.25 | 0.449±0.24 | 0.450±0.23 |
| UPNDOWN | 0.244±0.28 | 0.369±0.4 | 0.532±0.16 |

One observation from these results is that the myopic formulation performs worse in the $\ell_\infty$-norm bound than it does in the $\ell_1$-norm bound and the $\ell_2$-norm bound when $\epsilon = 10^{-8}$. A priori one might expect the performance of the myopic formulation to best in the $\ell_\infty$-norm bound because the unit $\ell_\infty$ ball contains the unit balls of the $\ell_1$ and $\ell_2$ norms. To investigate this further we examined the behaviour of $\ell_2$-norm bound and $\ell_\infty$-norm bound while varying $\epsilon$ in Figure (5). Observe that at very small scales the $\ell_2$-norm bounded perturbation has larger impact, but at larger scales the $\ell_\infty$-norm bounded perturbation generally performs better than the $\ell_2$-norm bounded perturbation. We link this phenomenon to the decision boundary behaviour in different scales.

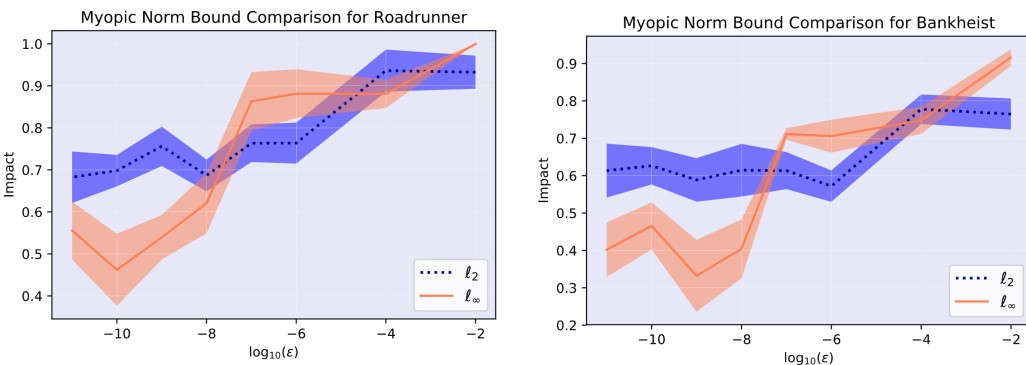

Figure 5: Left: Impact change with varying $\epsilon$ for $\ell_\infty$-norm bounded and $\ell_2$-norm bounded perturbation in Myopic formulation for Roadrunner game. Right: Impact change with varying $\epsilon$ for $\ell_\infty$-norm bounded and $\ell_2$-norm bounded perturbation in Myopic formulation for Bankheist game.

Recall that in the $\ell_2$-norm bound we use a perturbation in the gradient direction of length $\epsilon$. However, in the $\ell_\infty$-norm bound we use a perturbation given by $\epsilon$ times the sign of the gradient. This corresponds to the $\ell_\infty$-norm bounded perturbation which causes the maximum possible change for a linear function. Compared to previous formulations decreasing the temperature in the softmax makes our objective function more nonlinear. Thus, at smaller scales it is crucial for the perturbation to be in the exact direction of the gradient, since the decision boundary behaves nonlinearly. However, at larger scales the decision boundary is approximately linear, which gives a better result in the $\ell_\infty$-norm bound.

## A.2 RANDOMIZED ITERATIVE SEARCH AND IMPACTS

It can be seen from Figure 6 that the impact of the Pattanaik et al. (2018) formulation increases as $n$ increases, while our myopic formulation has a greater impact even when $n$ is equal to $1$. In particular, this is again an indication that our cost function provides a better direction in which to search for adversarial perturbations as compared to previous formulations.

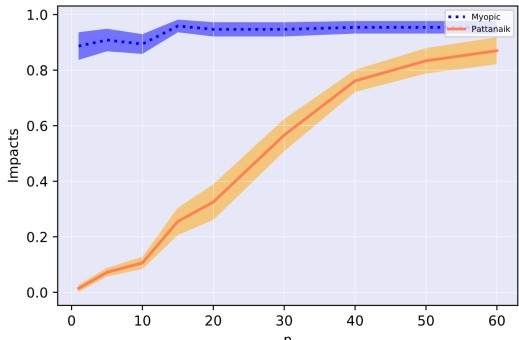

Figure 6: Attack impacts vs number of iteration of random search (n) for Pong.

Our attack, as well as those of Huang et al. (2017) and Pattanaik et al. (2018), are all based on computing the gradient of the respective cost functions and choosing a point along the gradient direction to be the adversarial perturbation. Pattanaik et al. (2018) used a randomized search, where $n$ random points are sampled in the gradient direction and the one with lowest $Q$-value is chosen as the perturbation. We follow Pattanaik et al. (2018) in using randomized search, and in all previous results mentioned above we have set $n = 5$. In Table 4 we set $n = 1$ and compare the impact of the three different formulations. Even in this more restrictive setting the performance of our formulation remains high, while the performance of Pattanaik et al. (2018) degrades significantly. In particular, this demonstrates that the gradient of our cost function is a better direction to find an adversarial perturbation.

Table 4: Attack impacts for three different attack formulations with $\ell_2$ norm bound and $\epsilon = 10^{-10}, n = 1$

| GAMES | HUANG | PATTANAIK | MYOPIC |
|---|---|---|---|
| AMIDAR | 0.050±0.25 | 0.138±0.31 | 0.941±0.02 |
| BANKHEIST | 0.189±0.13 | 0.247±0.15 | 0.487±0.09 |
| BEAMRIDER | 0.001±0.40 | 0.096±0.36 | 0.634±0.15 |
| RIVERRAID | 0.173±0.23 | 0.234±0.21 | 0.367±0.16 |
| ROADRUNNER | 0.035±0.15 | 0.090±0.12 | 0.151±0.12 |
| PONG | 0.173±0.23 | 0.014±0.03 | 0.887±0.09 |
| SEAQUEST | 0.321±0.14 | 0.290±0.4 | 0.502±0.22 |
| UPNDOWN | 0.475±0.24 | 0.615±0.10 | 0.911±0.04 |

## A.3 GAMES AND AGENT BEHAVIOUR

In this section we share our observations on the behaviour of the trained agent under attack. In Figure 9 the agent performs well until it suddenly decides to stand still and wait for the enemy to arrive. Similarly, in Figure 7 the trained agent performs quite well again until it decides to jump in front of the truck. Finally, in Figure 8 the trained agent forgets to recharge its oxygen even though it is earning many points from shooting the fishes and saving the divers.

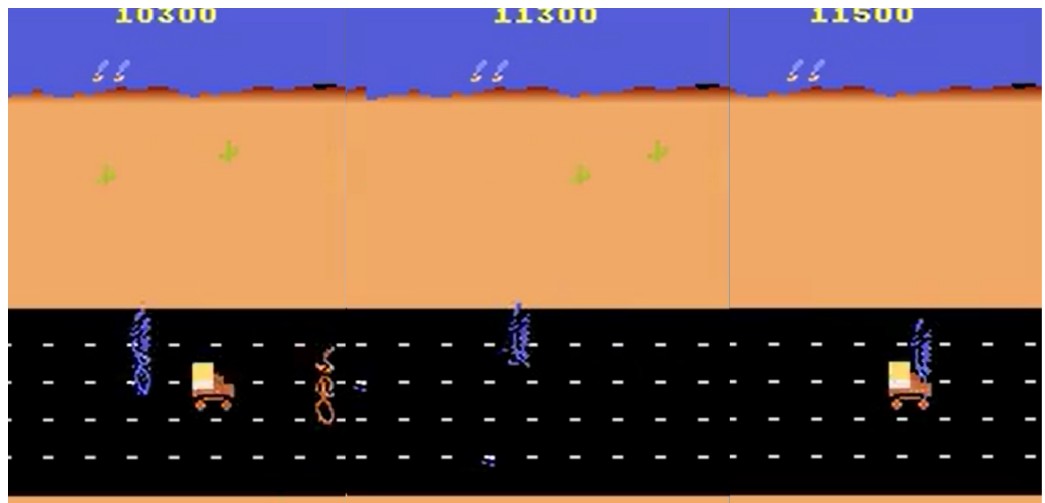

Figure 7: Example game from Atari Arcade Environment. The trained agent jumps in front of the car even though the agent is not in the same lane with the car in RoadRunner game.

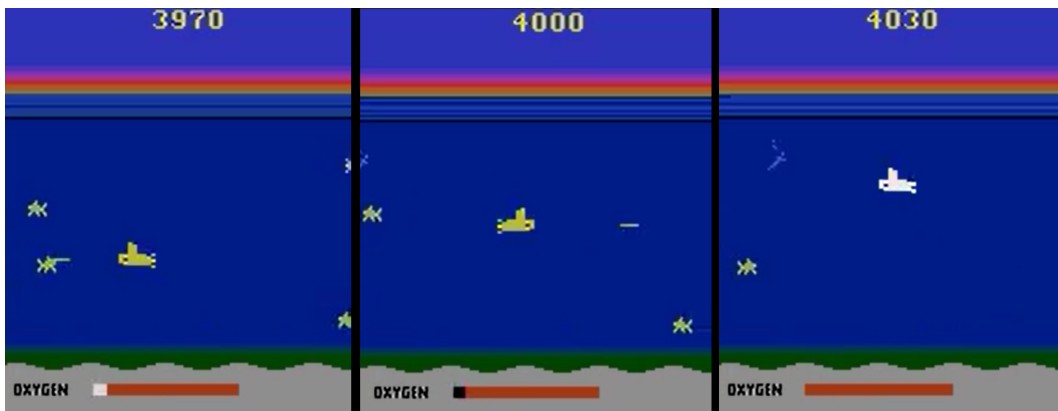

Figure 8: Example game from Atari Arcade Environment. The trained agent forgets to recharge its oxygen even though its condition to recharge is not critical in Seaquest game.

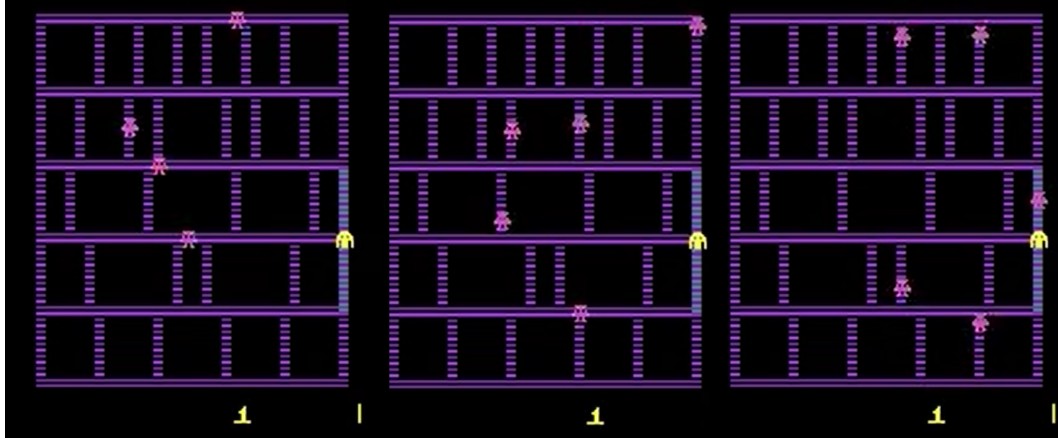

Figure 9: Example game from Atari Arcade Environment. The trained agent just waits without moving until the enemy reaches the agent in Amidar game.

### A.4    Q-VALUES OVER TIME

In this section we plot state-action values of the agents over time without an attack, under attack with the Pattanaik et al. (2018) formulation, and under attack with our myopic formulation. In Figure 12 it is interesting to observe that under attack the mean of the state-action values over the episodes might be higher despite the average return being lower. .

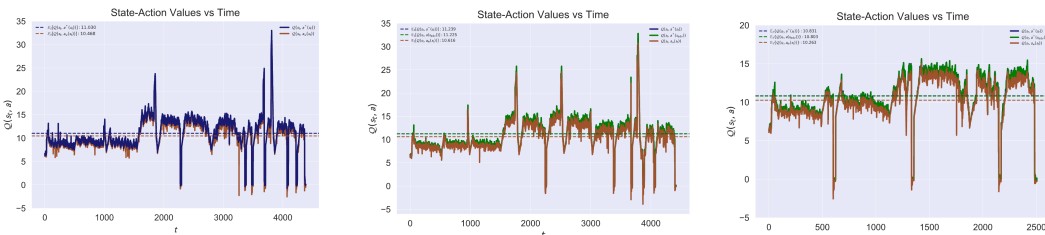

Figure 10: State-action values vs time graph for Atari Arcade Game Seaquest. Left: Unattacked agent. Middle: Pattanaik et al. (2018) adversarial formulation. Right: Myopic attack.

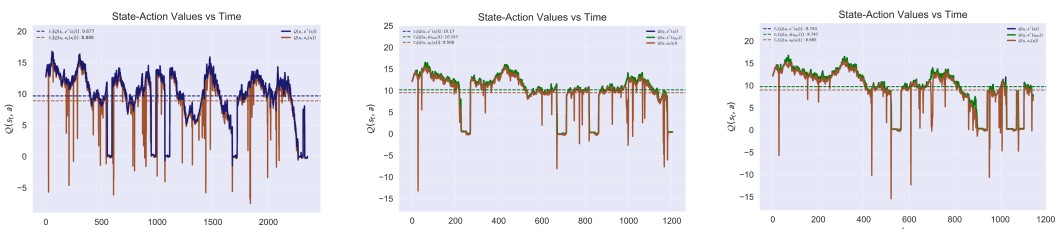

Figure 11: State-action values vs time graph for Atari Arcade Game Riverraid. Left: Unattacked agent. Middle: Pattanaik et al. (2018) adversarial formulation. Right: Myopic attack.

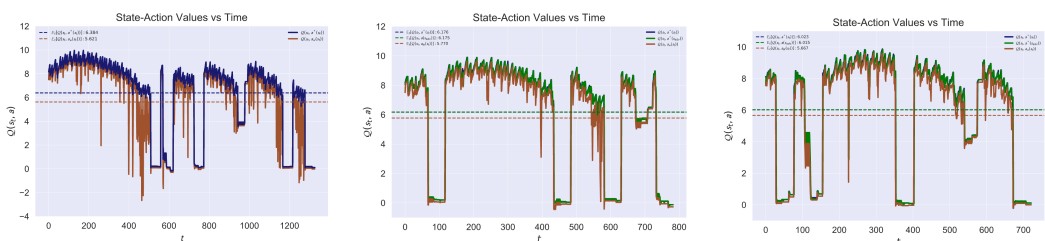

Figure 12: State-action values vs time graph for Atari Arcade Game Bankheist. Left: Unattacked agent. Middle: Pattanaik et al. (2018) adversarial formulation. Right: Myopic attack.

A.5 RESULTS ON AVERAGE EMPIRICAL PROBABILITIES OF ACTIONS

In this section we provide a detailed table on the average empirical probabilities of the best action $a^*$ and the worst action $a_w$ . It can be seen that in general $\mathbb{E}_{e\sim\rho}[p(s, a^*)]$ is lower and $\mathbb{E}_{e\sim\rho}[p(s, a_w)]$ is higher for our Myopic formulation when compared to other attack formulations.

Table 5: Average empirical probabilities of $a^*$ and $a_w$, and impacts for three different formulation for Riverraid and BeamRider from Atari environment with $\ell_2$-norm bounded perturbation and $\epsilon = 10^{-8}$.

|  | $\mathbb{E}_{e\sim\rho}[p(s, a^*)]$ | $\mathbb{E}_{e\sim\rho}[p(s, a_{\mathbf{w}})]$ | Impact |
|---|---|---|---|
| Riverraid |  |  |  |
| Huang et. al. | 0.9904±0.16 | 0.00226±0.0003 | -0.0796 ±0.19 |
| Pattanaik et. al. | 0.9443±0.29 | 0.00285 ±0.0014 | 0.3797±0.27 |
| Myopic | 0.897±0.16 | 0.00369±0.0012 | 0.5897±0.12 |
| BeamRider |  |  |  |
| Huang et. al. | 0.989±0.244 | 0.0042±0.0005 | 0.0847±0.29 |
| Pattanaik et. al. | 0.948±0.320 | 0.0059±0.0023 | 0.4553±0.267 |
| Myopic | 0.826±0.195 | 0.0186±0.0048 | 0.6636±0.135 |
| Amidar |  |  |  |
| Huang et. al. | 0.9872±0.168 | 0.0011±0.0008 | 0.04762±0.111 |
| Pattanaik et. al. | 0.9377±0.348 | 0.0052±0.0022 | 0.5145±0.126 |
| Myopic | 0.8566±0.211 | 0.01225±0.0035 | 0.9320±0.198 |
| BankHeist |  |  |  |
| Huang et. al. | 0.9861±0.103 | 0.000357±0.00059 | 0.111±0.14 |
| Pattanaik et. al. | 0.9426±0.012 | 0.00261±0.0122 | 0.373±0.15 |
| Myopic | 0.8974±0.122 | 0.00433±0.0036 | 0.624±0.08 |
| Pong |  |  |  |
| Huang et. al. | 0.4684±0.056 | 0.0020±0.001 | 0.0±0.016 |
| Pattanaik et. al. | 0.5151±0.096 | 0.01028 ±0.004 | 0.067±0.06 |
| Myopic | 0.5695±0.159 | 0.06221±0.013 | 0.920±0.08 |

## A.6 SENSITIVITY ANALYSIS

In this section we attempt to gain an insight into which pixels in the visited states are most sensitive to perturbation. For each pixel $i, j$ we measure the drop in the state action values when perturbing that pixel by a small amount. Formally, let $a_{\text{sen}} = \arg\max_a Q(s_{\text{sensitivity}}, a)$, where $s_{\text{sensitivity}}$ is equal to $s$ except that a small perturbation $\gamma$ has been added to the $i, j$-th pixel of $s$. Then we measure

$$Q(s, a^*) - Q(s, a_{\text{sen}}))$$ (19)

We plot the values from 19 in Figure 15. Note that any non-zero value (i.e. any lighter colored pixel in the plot) indicates that a $\gamma$ perturbation to the corresponding pixel will cause the agent to take an action different from the optimal one. Lighter colored pixels correspond to larger drops in Q-value.

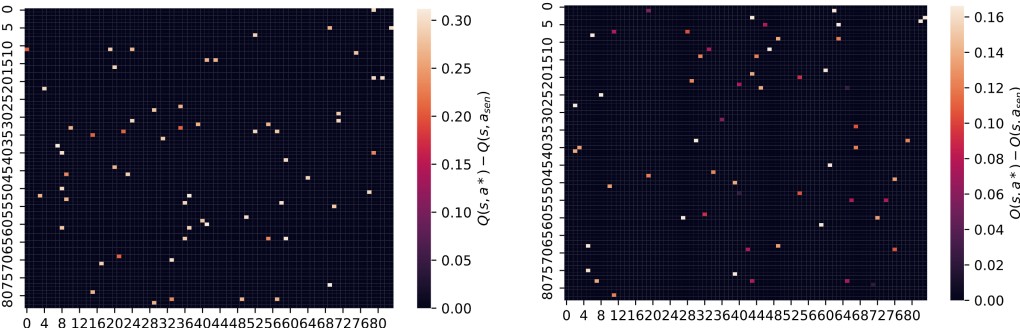

Figure 13: Sensitivity Analysis of RoadRunner. Left: $\gamma = 10^{-8}$. Right: $\gamma = 10^{-10}$.

## A.7 PERTURBATION HEATMAPS

In thise section we demonstrate the heatmaps of the raw pixel perturbations and the preprocessed frames from which the perturbations are computed.

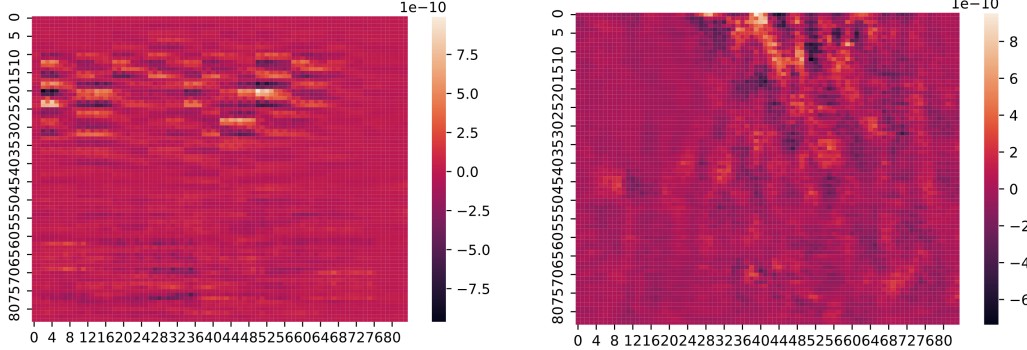

Figure 14: Perturbation heatmaps of myopic formulation. Left: RoadRunner. Right: Riverraid.

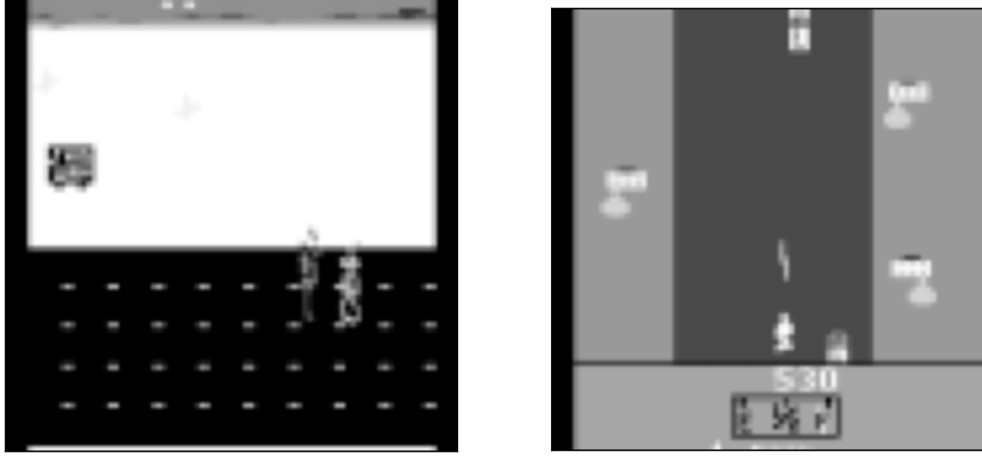

Figure 15: Preprocessed frames. Left: RoadRunner. Right: Riverraid.

