# OpenReview forum: "Bounded Myopic Adversaries for Deep Reinforcement Learning Agents"
_ICLR.cc/2021/Conference — Reject_

### Official Review · AnonReviewer1 · 2020-10-21
**Review of "Bounded myopic adversaries for deep reinforcement learning agents"**

**Rating:** 5
**Confidence:** 4

**Review:**

Summary: This paper proposes an optimal myopic adversary for deep reinforcement learning agent, in which the adversary finds a bounded perturbation of the state that minimizes the value of the action taken by the agent. The authors introduce a differentiable approximation for the optimal myopic adversarial formulation that leads to a better direction for the adversarial perturbation and increases the attack impact for bounded perturbations. Empirically, the authors show with experiments in various games in the Atari environment that the attack formulation achieves significantly larger impact as compared to the current state-of-the-art.

The paper is easy to follow, and the topic investigated looks interesting. However, a few major issues unfortunately in the draft discourage me to accept the paper.

1. The novelty in the paper is marginal. The conclusions drew from this paper are completely based on the experimental results. Though the formulation looks interesting, in-depth analysis is missing in the paper. No major theoretical results have been reported to provide stronger support for the bounded myopic adversary.

2. In Section 3, the authors have known that Eq (14) and Eq (15) may not be equivalent to each other, they fail to provide more analysis and discussion on how to address it, instead, only relying on the empirical results. Also, how to choose a sufficiently small adversarial temperature constant is not clear, as suggested in the paper. It gives a sense that the authors only focused on the applicability of the formulation, while ignoring necessary theoretical justification.

3. No empirical results for the continuous action set. The authors have mentioned the decent applicability of the myopic adversary in the continuous tasks. They should also have shown some results to support their claim, as failing to do this makes the paper look incomplete. Unless the proposed is only devoted to the discrete tasks.

Minor point:

4. Regarding the experimental results, I wonder why the authors put Huang and Pattanaik together for Atari games. Though this works, it would be better to see both discrete and continuous tasks separately for these two different baselines. Also, the evaluation metric in the paper is Impact instead of returns, which is not popular in literature. I suggest the authors to include this in the appendix.

*********************************
After carefully considering the rebuttal from the authors, I am going to maintain the score based on my evaluation and also the current paper draft. Though the authors have tried to addressed the comments, the paper still requires more improvements, including theoretical novelty and experimental results.

---

> ### Author Response · Authors · 2020-11-18
> **Author Response**
>
> Thank you for your comments. We tried to address them below.
>
> Our main goal for this paper was to show that the theoretical motivation for the optimal myopic adversary indeed gives state-of-the-art results for deep reinforcement learning in complex environments. We did this not only by showing the state-of-the-art impact results for our formulation, but by empirically verifying in Table 2 that the approximations we made in Equation (14) and Equation (15) were valid. More specifically, we intended the discussion leading up to Equation (14) and (15) to be used as motivation for our algorithmically tractable approximation to the optimal myopic adversary. We then verify empirically that the approximation is valid by demonstrating directly in Section 3.5 that the drop in Q-values in each state is larger for our formulation. Since this drop is exactly what the myopic adversary aims to achieve, this provides empirical evidence for the validity of the approximation beyond just showing the impact of our adversary on the agent’s final score.

---

> > ### Comment · AnonReviewer1 · 2020-11-23
> > **Thanks for the response**
> >
> > Thanks for your response!
> >
> > I completely understand the intention for the discussion between Eqs. (14) and (15). Though this might be acceptable in the work, still the theoretical novelty is marginal. As there is a gap between the formulation and the empirical results, which is the detailed analytical result for the proposed framework. The empirical results look promising in this paper, but without theoretical analysis, thorough empirical investigation that requires more different environments would be required, including the continuous action sets.

---

> > > ### Author Response · Authors · 2020-11-23
> > > **Thank you for your comments**
> > >
> > > Thank you for your valuable comments. We chose to evaluate our approach in a diverse set of environments in Atari because it is known to be a well-established, stable baseline. Doing an in-depth empirical analysis in these environments (i.e. in Section 3.3 - 3.5) allowed us to validate the assumptions made in our approach. We also believe that this is in line with prior work on adversarial deep reinforcement learning. In particular, Pinto et al. (2017) and Gleave et al. (2020), both focus solely on in-depth empirical analysis in one type of baseline to validate their approach. We believe that in-depth analysis in various environments from one baseline is a standard approach for this type of study. However, we of course agree with you that extending our approach to continuous action sets is interesting, and this is indeed a part of our future work. Thank you again for your response and your comments.
> > >
> > > Lerrel Pinto, James Davidson, Rahul Sukthankar, Abhinav Gupta: Robust Adversarial Reinforcement Learning. ICML 2017: 2817-2826
> > >
> > > Adam Gleave, Michael Dennis, Cody Wild, Neel Kant, Sergey Levine, Stuart Russell: Adversarial Policies: Attacking Deep Reinforcement Learning. ICLR 2020

---

### Official Review · AnonReviewer4 · 2020-10-26
**Strong empirical results, good intuitive justification, but threat model not justified and some assumptions/limitations unclear**

**Rating:** 6
**Confidence:** 4

**Review:**

The work introduces a new method for observation-perturbation adversarial attacks against deep neural network policies. The idealized version of the method is an optimal attack assuming (a) the agent's Q-values are calibrated; (b) the attacker can only act once, at the current time step. While the attack is in fact conducted at every time step, so is not "myopic", this assumption greatly simplifies the problem. Moreover, the work approximates the (deterministic, hard-max) policy with a low-temperature softmax policy to enable gradient-based methods to work. Evaluating in a range of Atari games, the method has a significantly larger "impact" (normalized return) than previous methods, across $\ell_1$, $\ell_2$ and $\ell_{\infty}$ norms. Moreover, the results suggest the attack succeeds because of increasing the probability of low-ranked actions, consistent with the attack's derivation.

Strengths:
  - Empirical results show a substantial improvement over a reasonable choice of baseline methods.
  - Experiments also support author's claims for why the method works (especially section 3.4 but also to a lesser extent section 3.5) -- important given the approximations needed from the derivation.
  - The paper gives a good intuition for why the method should work, and clearly highlights the limitations of previous methods.

Weaknesses:
  - The justification of the threat model could be improved. I appreciate the list of examples in paragraph 3 of the intro for where RL applications have been studied, but why is this threat model actually relevant to these example? For example, take financial trading: an attacker cannot perturb other peoples orders in the market data feed (unless they've compromised the exchange -- in which case there are much simpler routes to exploitation!), but can insert their own orders with almost arbitrary price/size. This does not fit an $\ell_p$ norm, which assumes modifying any field is equally possible. I would suggest picking a use case where this threat model does hold and going into detail on it -- or if there is none, then be more up front about this limitation.
  - Contrary to section 4 I am not convinced this method can be easily applied to continuous control tasks. First, policies for continuous control need not be stochastic. Typically, the policy outputs parameters of a distribution. At training one samples from this distribution -- but at inference one often takes the mode, deterministically, since it improves performance. Even if we allow for a stochastic policy, one has the new challenge of a much larger action space.

    My best guess is the method could be made to work, given that e.g. Pattanaik's method is not all that different and worked on MuJoCo. But I expect it to require some non-trivial tuning, and possibly new algorithmic insights. This section should really be supported by experiment, or have the claim reduced in scope to there being no *obvious theoretical hurdles* to its application.
  - Paper should be more up-front about assumptions being made. I did like the last paragraph of section 3 (before section 3.1). But you make other assumptions implicitly. For example, the agent's Q-values may be miscalibrated, such that the worst action for the agent is not actually the one the agent assigns lowest Q-value to! This will make your method suboptimal (even in the myopic case). Additionally, the fact you attack at each time step but choose the attack myopically must be leaving some value on the table. Indeed, the fact that expected Q-values often rise (table 2) under the attack is some evidence of this. I think the paper is a good submission without fixing these limitations (though it would be much stronger if they were addressed), but it's important they're clearly signposted so that they can be addressed in future work.

I consider this paper borderline but overall am leaning towards accept. There is a clear theoretical reason why the idealized method should perform better than previous methods (e.g. Huang, Pattaniak) and the empirical results support that the approximation is stronger than baselines. The threat model of this (and prior work) seems chosen more for mathematical simplicity than realism, but may help lay the ground for future work in more realistic settings. The paper could do a better job of communicating the assumptions and limitations, but assuming this is addressed then it seems worth disseminating to the ICLR community.

A few questions:
  - Do you have any additional insights into why the mean Q-value per episode often increases under attack? This is a very counterintuitive result -- why should choosing worse actions lead you to higher-valued states? Some things that might be worth looking into (though I certainly do not expect all of these during the relatively brief discussion period):
    + What return do you get if you perform the attack for 1-timestep (or a smaller number of timesteps) and then run the unattacked agent? This is what the Q-value is actually estimating -- but Q-values learned by deep RL are often very uncalibrated, especially off-distribution.
    + What happens in other environments? It strikes me that most Atari games are quite hard to get "stuck" in: even if you die you respawn. Some environments usually used to test safe exploration might be helpful with this.
    + What happens with attacks that are non-myopic, e.g. the "enchanting attack" of Lin et al (2017)?
  - Do you have any experiments in continuous control tasks that would back up section 4, even if preliminary?
  - Why are \ell_p norm perturbations of observations an important threat model to study? I know it is widely studied, but it is also widely criticized -- e.g. Gilmer et al (2018) -- and for RL in particular it seems rare for an attacker to be able to directly perturb observations. Are there cases where this attack is realistic? Are there non-security reasons to care, e.g. to improve robustness to natural "perturbations"? If so, can we validate this in a realistic setting, e.g. does vulnerability to your method also predict failures of sim2real transfer, or sensitivity to non-adversarial noise?
  - What do the perturbations actually look like? Do they look like anything reasonable, e.g. introducing a fake "ball"? Given you claim the perturbation direction identified by this method is likely stronger, it'd be interesting to know what neural networks find most "persuasive".

Detailed feedback:
  - Need to use parenthetical citations \citep in places rather than in-line citations \citet: e.g. first paragraph in the intro “speech recognition Hannun et al. (2014)” -> “speech recognition (Hannun et al, 2014)”; first paragraph of 3.1.
  - “perturbations to image”->”perturbations to images”.
  - Intro: focus it more on your method. It's a good summary of some of the history of adversarial examples – but is this that relevant? Most readers (and certainly your reviewers) will be familiar with adversarial examples. Adversarial examples in RL might need some handholding being less common, but can probably also be assumed.
  - Related, could benefit from an explicit statement of what your threat model is early on (an adversary making bounded perturbations of the observations of an RL agent). I appreciate the list of diverse scenarios RL has been applied it, but I would suggest making this more focused on those that are actually deployed on nearing deployment (especially if safety critical).
  - Related Work: 2.1 is a nice succinct summary of existing work but it'd be helpful to the reader to place it in relation to your own work. In particular it seems your method is much more closely related to Huang, Kos & Song and Pattanaik et al than it is to Gleave et al or Pinto et al. It therefore might be useful to treat these as two groups (“multi-agent” adversarial RL v.s. “observation” adversarial RL, say); discuss why you chose to work under the observation threat model; and then dig into details on how your approach differs from Huang, Kos, Pattanaik (e.g. is it the same threat model but stronger empirical results? do you make stronger/weaker assumptions?).
  - “$\pi(s,a)$ is not the actual policy used by the agent” – bit confusing, can you change notation to make this explicit e.g. $\pi_{\text{soft}}$? Also good to be clear the difference is just soft vs hard-max (I assume), rather than e.g. $\pi$ being learned by behavioral cloning the policy under attack. Perhaps explicitly state earlier your attack is white-box?
  - I liked section 2.3: it was an easy to read summary that clearly explains the problems with existing work.
  - Section 3: your method is not really “optimal” since you approximate the hard-max policy with a soft-max policy of low temperature, so the section is a bit misleading. It might be better to frame it as: (1) definition of optimal attack; (2) tractable approximation to this. I do appreciate the discussion in the last paragraph of section 3 (before section 3.1) of why this is an approximation.
  - Figure 4: can you use the subcaption package to give each individual barchart its own label? This is easier than having to consult the main caption. It's also worth making explicit that these plots do not inculde the best action – I was wondering for a while where the rest of the probability mass was! I'd be inclined to even add the best action to the plot (perhaps visually distinguished in some way, and using a log-scale for probabilities if it would compress it too much).
  - Table 2: caption would benefit from being expanded.

Update after author response: The changes made have improved the clarity of the paper, such as making assumptions and the threat model more explicit, and the heatmap addition provides a nice qualitative insight. However, I am inclined to agree with other reviewers that the paper's contribution is incremental. Given this I am retaining my score of marginally above acceptance.

---

> ### Author Response · Authors · 2020-11-18
> **Author Response Part II**
>
> 3. "Why are \ell_p norm perturbations of observations an important threat model to study?... Are there non-security reasons to care, e.g. to improve robustness to natural "perturbations"? ":
>
> There are non-security reasons to care about \ell_p norm perturbations. In adversarial training norm bounded perturbations are used at training time to improve robustness. It has been shown in this setting (Madry et al 18) that it is important to have the strongest possible norm bounded attacks in order to ensure that adversarial training performs well. Further, work has also shown that adversarially trained image classifiers internal features are useful out-of-the-box for transfer learning tasks (Salman et al 2020).
>
> Aleksander Madry, Aleksandar Makelov, Ludwig Schmidt, Dimitris Tsipras, Adrian Vladu: Towards Deep Learning Models Resistant to Adversarial Attacks. ICLR (Poster) 2018
>
> Hadi Salman, Andrew Ilyas, Logan Engstrom, Ashish Kapoor, Aleksander Mądry: Do Adversarially Robust ImageNet Models Transfer Better?. NeurIPS 2020.
>
>
>  4. "The claim reduced in scope to there being no obvious theoretical hurdles to its application.":
>
> In Section 4 our aim was to give an insight into how it might work in continuous tasks.  As you suggested we just wanted to claim there wouldn’t be any obvious theoretical hurdles in it’s application. We rephrased this section. Please let us know if you have anything more you wanted us to add on how to explain the continuous control tasks.
>
>
> 5. "Why the mean Q-value per episode often increases under attack?":
>
> In Section 3.5 we wanted to emphasize that when the attack is applied in every time step, the key assumption made by Q-learning (i.e. that the action taken in future states will maximize the Q-value) is violated. As a result we expect the Q-value estimates to be wrong, either by being higher or lower. Note that in some games (Riverraid, Seaquest, Amidar) Myopic gets higher average Q-values than the unattacked agent, but in Bankheist the average Q-value is lower. In other words there is not a consistent trend across games in the direction in which the Q-value estimation errs under attack.
>
> We hypothesize that the success of our attack lies in causing the agent to take very bad actions at key states (e.g. being killed by an enemy, missing the ball in pong). This corresponds to having several states where the $Q(s,a_{adv})$ is much lower than $Q(s,a^*)$. This drop is precisely what is measured by the $Q_{loss}$ metric which we show is consistently larger across games for our attack when compared to others.
>
> 6. "What do the perturbations actually look like? Do they look like anything reasonable, e.g. introducing a fake "ball"? Given you claim the perturbation direction identified by this method is likely stronger, it'd be interesting to know what neural networks find most "persuasive".":
>
> This is a very exciting question to ask. We also added heatmaps of the adversarial perturbations in the Appendix Section A.7.
>
> Thank you so much for your time and for your detailed feedback. We tried to address them all and revised the paper. Please let us know if you have any more comments to add. We would be very happy to revise it.

---

> > ### Comment · AnonReviewer4 · 2020-11-20
> > **Thanks for the clarification**
> >
> > Thank you for your detailed response to my questions.
> >
> > > For the financial trading example you are right that some of the  norm bounded perturbations do not really make sense i.e. . However, does actually make sense due to the sparsity of the  perturbations produced by our attack.
> >
> > I appreciate the elaboration you provide on this example, and I agree that the $\ell_1$ norm could be a reasonable model for attacks on financial markets when the RL system observes a price-quantity order book. My suggestion would be to pick one or two examples (it needn't be this) and do a similarly deep analysis in the intro. I suspect most readers will be aware that researchers have applied deep RL in a variety of fields, but may have less of a handle on what a realistic attack in an RL setting looks like. You have an extra page for the rebuttal & camera-ready so you can keep the wide list of applications, though I'd suggest only including those where you consider an attack to be actually plausible.
> >
> > > We have added a paragraph before Section 3.1 to incorporate them.
> >
> > Thanks for adding this paragraph, this will help inform the reader of possible limitations and largely alleviates my concern in that regard.
> >
> > > We rephrased this section. Please let us know if you have anything more you wanted us to add on how to explain the continuous control tasks.
> >
> > Thanks, I think this is sufficient. Of course experiments in continuous control tasks would be the most compelling, as other reviewers mentioned, but I appreciate there is unlikely to be time for this during the discussion period.
> >
> > Typo: "actually it is easier" -> "it is actually easier"
> >
> > > In Section 3.5 we wanted to emphasize that when the attack is applied in every time step, the key assumption made by Q-learning (i.e. that the action taken in future states will maximize the Q-value) is violated. As a result we expect the Q-value estimates to be wrong, either by being higher or lower.
> >
> > I don't fully understand this explanation. $Q(s,a^*(s)) = V(s)$, and the policy (not attacked) should want to go to states with the highest value. So the fact that you get a higher mean value when you perform the attack in three out of four environments seems surprising. I could understand this if your method was systematically searching for cases where the Q-value is wrong, but you're just trying to force the agent to choose what is (by its own lights) the worst action at each time step. So why should the agent being more likely to take the worst action lead to states with higher value?
> >
> > That said, this seems to occur with the baseline methods too (although is more marked in your method), so is not something that the paper needs to explain.
> >
> > > This is a very exciting question to ask. We also added heatmaps of the adversarial perturbations in the Appendix Section A.7.
> >
> > Thanks. One typo: "hetmaps" -> "heatmaps". It might be helpful to also show the raw video frame the perturbations are being applied to? In particular, after any preprocessing (rescaling, greyscaling) has been applied so we can see what the RL agent sees. I'm a little confused by the perturbations right now -- for RoadRunner it looks like it's adding dashes to the sky? Perhaps that confuses the agent as to where the road is? The RiverRaid one makes more sense, adding perturbations to the horizon the agent is approaching, though there's nothing recognizable.

---

> > > ### Author Response · Authors · 2020-11-23
> > > **Thank you for the insightful review**
> > >
> > > 1. Thank you for your suggestion on the intro and specific adversarial scenario. We have added a particular scenario where our attack might be of interest.
> > >
> > > 2. Thank you for pointing out the typos. We have fixed them too.
> > >
> > > 3.  This is a great idea. We have also added the raw video frames in Appendix A.7. It is quite interesting that the RoadRunner perturbation is primarily modifying the sky, an area of the frame which is not affected by the agent's actions. One possible explanation is that in training the sky remains more or less uniform, so a perturbation which causes very different features to appear in the sky is somehow very off-distribution for the trained agent. It also does look suspiciously like the road as you point out.

---

> > > > ### Comment · AnonReviewer4 · 2020-11-25
> > > > **Revision**
> > > >
> > > > Thanks for the revisions. Modifications to the sky being highly off-distribution is a plausible explanation.

---

> ### Author Response · Authors · 2020-11-18
> **Author Response Part I**
>
> Thank you for your detailed comments and questions. We tried to address them below.
>
> 1."For example, take financial trading: an attacker cannot perturb other peoples orders in the market data feed (unless they've compromised the exchange -- in which case there are much simpler routes to exploitation!), but can insert their own orders with almost arbitrary price/size. This does not fit an ℓp norm, which assumes modifying any field is equally possible. I would suggest picking a use case where this threat model does hold and going into detail on it -- or if there is none, then be more up front about this limitation.":
>
> For the financial trading example you are right that some of the $\ell_p$ norm bounded perturbations do not really make sense i.e. $\ell_\infty$. However, $\ell_1$ does actually make sense due to the sparsity of the $\ell_1$ perturbations produced by our attack. If we consider  a model with some large set of financial traders in a market, due to the $\ell_1$ sparsity (Candes et al. 2006) we only need a much smaller subset of adversarial traders to compromise the whole trading system.
> Also while it is true that inserting orders with arbitrary price or size is possible for an adversary, such activity might trigger automatic anomaly detection schemes, thus compromising such an attack. In other words, this can be seen as a more detectable type of manipulation to the system than an extremely small, sparse perturbation (e.g. $\ell_1$-norm). In this case smaller perturbations that still succeed may be less likely to be detected and less risky for the adversary. Thus it is important to identify and investigate $\ell_p$-norm attacks also from the security point of view.
>
> More than providing a specific security application we wanted to give an overview on where the algorithms are currently being deployed in real life. To be precise, we wanted to connect the deployment of these models with their vulnerability to small perturbations, and how much we need them to be robust in these settings. However, we agree perhaps we couldn’t really convey this as it is written now. We could remove some of the applications if you feel they do not fit the vulnerabilities that we discuss in our paper.
>
> Emmanuel J. Candès, Justin K. Romberg, Terence Tao: Robust uncertainty principles: exact signal reconstruction from highly incomplete frequency information. IEEE Trans. Inf. Theory 52(2): 489-509 (2006).
>
>
> 2." Paper should be more up-front about assumptions being made. I did like the last paragraph of section 3 (before section 3.1). But you make other assumptions implicitly. For example, the agent's Q-values may be miscalibrated, such that the worst action for the agent is not actually the one the agent assigns lowest Q-value to! This will make your method suboptimal (even in the myopic case). Additionally, the fact you attack at each time step but choose the attack myopically must be leaving some value on the table. Indeed, the fact that expected Q-values often rise (table 2) under the attack is some evidence of this. I think the paper is a good submission without fixing these limitations (though it would be much stronger if they were addressed), but it's important they're clearly signposted so that they can be addressed in future work.":
>
> Yes, you are exactly right. In a myopic formulation some value definitely is left on the table. Because the action that minimizes the state-action value distribution in the current state assumes that the agent is going to take the action maximizing its state-action value distribution in the following states, without the influence of the adversary. This is definitely a great question to answer. Our future and follow-up work is partially focused on extending the myopic formulation on this basis. Note however that non-myopic attacks such as Lin et al (2017) require training a generative model for future state predictions whereas the optimal myopic adversary does not require any training in the agent’s MDP. Thus, in a sense the optimal myopic adversary is operating in a more restricted threat model.
>
> It might be reasonable to be concerned about the case where the Q-values are miscalibrated. In such a case it is not clear how helpful it will be to myopically minimize the Q-value. Our experiments showing that our attack achieves higher impact and makes the agent take lower ranked actions are one form of evidence that the Q-values are well calibrated enough to make our approach effective. As additional evidence of this, we observed that forcing the agent to take the lowest ranked action in each state often gives scores that are lower than those of an agent taking uniformly random actions.  Thank you for your well-thought out comments on these nuanced points. We have added a paragraph before Section 3.1 to incorporate them.

---

### Official Review · AnonReviewer3 · 2020-10-29
**Techniques presented could improve analysis of perturbations needed to efficiently achieve significant impacts on DeepRL agents, but the experimental results need further explanation**

**Rating:** 6
**Confidence:** 3

**Review:**

Summary:  This paper proposes to build an adversary to find a bounded perturbation of the state that minimizes the value of the action taken by the reinforcement learning agent. This approach enables us to lower the bounds by several orders of magnitude on the perturbation needed to efficiently achieve significant impacts on DeepRL agents.

Strong aspects:
1.     The approach is simple and straightforward with good numerical performance.
2.     The writing is easy to follow and experiments are thorough.

Weak aspects:
1.     It is strange that, for a state of thousands of dimensions, a tiny perturbation within a ball having a radius of $10^-10$, can make maximum possible deterioration in terms of return (i.e., Fig. 3). Can the authors provide more insights? For example, the sensitivity analysis of the return / Q value with regard to the states might help.

2.     How does the proposed approach compare to other more recent baselines such as Gleave et al. (2020)?

3.     Is the metric in (16) a common one? What if we simply plot the return under various perturbations, like that in Huang et al. (2017)?

Minor points:

- The locations and / or sizes of the legends in Fig. 2, 3 and 4 and the size of axis labels can be adjusted accordingly to make them easier to read.


I have read the authors' response and the associated discussions, and based on that  raised my evaluation by 1

---

> ### Author Response · Authors · 2020-11-18
> **Author Response**
>
> Thank you for your comments and suggestions. We have tried to address them below.
>
> 1."It is strange that, for a state of thousands of dimensions, a tiny perturbation within a ball having a radius of $10^{-10}$, can make maximum possible deterioration in terms of return (i.e., Fig. 3). Can the authors provide more insights? For example, the sensitivity analysis of the return / Q value with regard to the states might help.":
>
> We added a sensitivity analysis in Appendix Section A.6 demonstrating that even changing one pixel by $10^{-10}$ can cause the agent to take a non-optimal action. In that section we include plots showing, for a given state, the set of pixels that have this property.
>
> 2. "How does the proposed approach compare to other more recent baselines such as Gleave et al. (2020)?"
>
> Gleave et al. (2020) is based on a multi-agent setup where one agent is controlled by the adversary, but restricted to taking only legal actions in the environment. They show that by taking highly out-of-distribution actions in a multi-agent environment the adversary agent can cause the victim agent to fail. So in this setup the adversary’s out-of-distribution behavior is in a sense quite perceptible, but is restricted to only legal actions within the environment.
>
> In contrast, in our setup the attack is an imperceptible perturbation, restricted to be in some small $\ell_p$-norm bounded ball. In this sense these two different approaches are incomparable, while both testing interesting aspects of adversarial vulnerability.
>
> 3. "Is the metric in (16) a common one? What if we simply plot the return under various perturbations, like that in Huang et al. (2017)?":
>
> We reported the results as impacts because we wanted to give a normalized scale so it would be more reasonable to compare across different games and adversarial formulations. If you have any suggestions on different ways to normalize the scores we would be happy to report them in that way.
>
> Thank you for your comments and questions. We have tried to address them. Please let us know if you have more questions. We would be happy to answer them.

---

### Official Review · AnonReviewer2 · 2020-10-29
**The paper has a nice idea but a little incremental**

**Rating:** 6
**Confidence:** 4

**Review:**

This paper proposes using a better quantitative metric to conduct an attack on a DRL learner. The attack is limited to an attack on a state (not over multiple states) and aims to lower the Q value from this state by making the worst action be the action chosen to be played. The experiment on Atari games show promising results.

Pros:
- The idea is effective, finding the right attack objective is interesting and surprising that was not considered earlier.
- Good description of why prior methods do not achieve the optimal attack
- The experiments show good results

Cons:
- The techniques are not very novel, softmax (with temperature) as a soft differentiable version of argmax is very well known.
- A comparison to non-myopic attack would make paper stronger (https://arxiv.org/pdf/1907.09470.pdf)
- The legends in the figure are just too small to be readable
- Would have been good to show attacks on more complex problems.

Questions:
- This attack is for a particular state, which state is chosen for this attack? Is it towards the start of the game or end of the game?
- Why is E[Q(s; aw)] different for different approaches in Table 2, and also for E[Q(s; a*(s))]? These values should not depend on the attack.
- Aren't the variances too high in Table 1? Is 10 episodes enough - why not more?

---

> ### Author Response · Authors · 2020-11-18
> **Author Response**
>
> Thank you for your comments and suggestions. We tried to address them below.
>
> 1. "This attack is for a particular state, which state is chosen for this attack? Is it towards the start of the game or end of the game?":
>
> The attack is not applied to a single state, but rather myopically in each state. In other words, for each state visited by the agent the attack attempts to minimize the Q-value of the action taken in that particular state.
>
> 2. "Why is E[Q(s; aw)] different for different approaches in Table 2, and also for E[Q(s; a*(s))]? These values should not depend on the attack.":
>
> $E[Q(s; a_w)]$ and $E[Q(s; a^*(s))]$ actually do depend on the adversarial attack, because the agent visits a different set of states depending on which attack is used. The values $E[Q(s; a_w)]$ and $E[Q(s; a^*(s))]$ are the state-action values of the worst and best actions averaged over the states visited by the agent. Since the agent visits different sets of states for the different attacks, these values end up being different. This can be seen in Figure 4 which shows the action distributions of the agents with different adversarial formulations.
>
> 3. "Aren't the variances too high in Table 1? Is 10 episodes enough - why not more?":
>
> It is important to note that the standard deviations are higher for the prior work and significantly smaller for the optimal myopic adversary. The reason we picked 10 episodes was to stop once we had reasonably stable data. Additionally, greatly extending the number of episodes in order to achieve a consistent impact can also be seen as a weakness of the adversary from a security point of view. Also note that the standard deviations are small enough to clearly distinguish our attack from the prior work (for a visual representation of the confidence intervals see for example Figure 3).
>
> 4. "Would have been good to show attacks on more complex problems.":
>
> Could you please elaborate on what is meant by more complex problems? We focused on the Atari environment because it is an established baseline providing a fair comparison between different training algorithms and adversarial formulations. Also it is important to note that even well established and well-performing algorithms like DDQN do not have uniformly good performance across all Atari games, but may do poorly in several of them. In this sense these baselines are not trivial for established algorithms like DDQN, and so demonstrating attacks in semantically different games in this baseline is a reasonable approach to test a new attack method. Of course we do agree that the future work could perhaps consider real-life problems in realistic settings. However, we also think the follow-up work can only proceed after firstly showing that the proposed algorithms work in the established baselines.
>
> 5. "The techniques are not very novel, softmax (with temperature) as a soft differentiable version of argmax is very well known.":
>
> We believe that the main novelty in the paper is in introducing a new formulation which directly maximizes the drop in Q-values caused by the adversary, unlike the approaches in prior work. In the description of the use of softmax as a differentiable approximation to argmax we wanted to give an insightful thorough compact formulation of the approximation for the optimal myopic adversarial policy. We are willing to rephrase this technical detail  in a way that is more compact. Please let us know if you think rephrasing this description would improve the clarity of the paper.
>
> 6. “The legends in the figure are just too small to be readable”:
>
> Thank you for the suggestion. We fixed them in the paper.
>
> Please let us know if you have any further comments. We would gladly try to address them.

---

> > ### Comment · AnonReviewer2 · 2020-11-21
> > **More explanation**
> >
> > Thank you for the clarifications.
> >
> > A few points are still not clear to me.
> >
> > Can you please in more details explain what do you mean by E[Q(s; a*(s))] - I do not get why this depends on the attack? Is the best action not being taken in every state? What is the expectation over?
> >
> > If the attack in in every state, then I feel comparison to a non-myopic attack is even more required. (https://arxiv.org/pdf/1907.09470.pdf)
> >
> > What do the authors mean by this "The reason we picked 10 episodes was to stop once we had reasonably stable data."? What does reasonably stable mean? I just think that averaging over more episodes will reduce variance and provide more correct expected values even for previous approaches - clearly, for beamrider, roadrunner (the ones not shown visually) the variance leads to overlap.
> > Also, I am not sure I understand the claim that greatly extending the number of episodes in order to achieve a consistent impact can also be seen as a weakness of the adversary from a security point of view. Can the authors please explain why?

---

> > > ### Author Response · Authors · 2020-11-22
> > > **Additional Explanation**
> > >
> > > 1. "Can you please in more details explain what do you mean by E[Q(s; a*(s))] - I do not get why this depends on the attack? Is the best action not being taken in every state? What is the expectation over?":
> > >
> > > When under attack the agent indeed takes the best action in the **“observed”** state. Due to the attack, the state observed by the agent is the perturbed state $s_{adv}$, while the true visited state is $s$. Thus the agent takes the action $a_{adv}$ maximizing $Q(s_{adv},a)$, which is different than the action $a^*$ maximizing $Q(s,a)$. The expectation $E[Q(s,a^*)]$ is the expectation over true visited states $s$. This means that for each state visited by the agent, we compute $Q(s,a^*)$, the value of the best action in the true visited state, and then average this value over all visited states in the episode. Since the different attacks will cause a different distribution over states $s$ visited by the agent, this value will be different for each attack.
> > >
> > > 2. "What do the authors mean by this "The reason we picked 10 episodes was to stop once we had reasonably stable data."? What does reasonably stable mean? I just think that averaging over more episodes will reduce variance and provide more correct expected values even for previous approaches - clearly, for beamrider, roadrunner (the ones not shown visually) the variance leads to overlap. Also, I am not sure I understand the claim that greatly extending the number of episodes in order to achieve a consistent impact can also be seen as a weakness of the adversary from a security point of view. Can the authors please explain why?":
> > >
> > > Each standard deviation listed in the table is the sample standard deviation $s$ i.e. it is an estimate of the true population standard deviation $\sigma$ of the distribution on impacts of the adversary. The quantity $s$ of course does not go down with more samples, but rather becomes a more accurate estimate of the true standard deviation $\sigma$ of the distribution of impacts.
> > >
> > > To show that our formulation has a higher impact on average, the correct metric is the standard error of the sample mean $\sigma_{\bar{x}} = \sigma/\sqrt{n}$ where $n$ is the number of samples. As is standard we estimate $\sigma_{\bar{x}} \approx s/\sqrt{n}$ using the sample standard deviation $s$. Using this estimate for the standard error of the mean, we see that we do indeed have high confidence that our average impact is higher. Please see below for a table reporting the standard error of the mean.
> > >
> > > The reason we reported the sample standard deviation in Table 1 in the paper is to give some quantitative information on the shape of the distribution on impacts achieved by each attack. Our attack not only has higher mean impact, but also has a distribution on impact which is more tightly concentrated around its mean.
> > >
> > >
> > >
> > > Table A: Standard error of the mean Impact.
> > >
> > > | Games      | Huang et al. (2017) | Pattanaik et al. (2018) |   Myopic    |
> > > |------------|---------------------|-------------------------|-------------|
> > > | Amidar     | 0.048±0.06          | 0.514±0.092             | 0.932±0.006 |
> > > | BankHeist  | 0.111±0.044         | 0.373±0.047             | 0.624±0.025 |
> > > | BeamRider  | 0.083±0.095         | 0.455±0.085             | 0.663±0.041 |
> > > | RiverRaid  | -0.079±0.060        | 0.379±0.085             | 0.589±0.037 |
> > > | RoadRunner | 0.187±0.050         | 0.387±0.050             | 0.557±0.034 |
> > > | Pong       | 0.0±0.005           | 0.067±0.018             | 0.920±0.025 |
> > > | Seaquest   | 0.305±0.082         | 0.524±0.072             | 0.697±0.050 |
> > > | UpNDown    | 0.074±0.095         | 0.476±0.050             | 0.865±0.019 |
> > >
> > >
> > >
> > > 3. "If the attack in in every state, then I feel comparison to a non-myopic attack is even more required. (https://arxiv.org/pdf/1907.09470.pdf)":
> > >
> > > Could you please specify which attack you would like to see a comparison to? In the paper you reference the best performing attack in the Atari environment seems to be “obs-fgsm-wb” which is precisely the attack from Huang et al. (2017). We do indeed compare to this attack as you can see in our paper in Table 1, Table 2 and Figure 3.
> > >
> > > We were not aware of this paper, as it is an unpublished manuscript. However, since these other attacks don’t seem to perform better than Huang et al. (2017) it is not exactly clear to us that there is some other baseline comparison to make with the linked manuscript.

---

### Author Response · Authors · 2020-11-25
**Author Meta Response**

Dear Area Chairs and Reviewers,

We thank the reviewers for finding our approach effective, intuitive, interesting, straightforward and surprising, and finding our experiments thorough.

Thank you all for your effort and time in the reviewing process. We responded to each reviewer individually. Here we write the summary of the author rebuttal process:

* **[18.11.2020]** Sensitivity analysis has been added in Appendix Section A.6.

* **[18.11.2020]** Adversarial perturbations are added in Appendix Section A.7.

* **[18.11.2020]** More explicit discussion on the assumptions for myopic formulations has been added in Section 3.1.

* **[18.11.2020]** The introduction of Section 4 has been rephrased.

* **[23.11.2020]** Specific motivating adversarial scenario has been added in Introduction.

---

### Decision · Program_Chairs · 2021-01-07
**Final Decision**

**Decision:**

Reject

**Comment:**

Most reviewers are positive about this work, though they believe it is somewhat incremental, and its theoretical contributions are minor. None of the reviewers are very excited about this work. Overall, the PC believes this is a borderline paper.

Minor note: During the discussions, the paper by Xiao et al., "Characterizing Attacks on Deep Reinforcement Learning" (2019) was brought up. The authors claimed that they did not compare with that paper because the best attack there (obs-fgsm-wb) had already been studied. In a later stage of discussions, one of the reviewers stated that the method obs-nn-wb in that paper performed better in some domains. Even though this is not a major issue, it is advisable to the authors to make sure that this is indeed the case, and if it is, provide proper comparison with that paper.

We encourage the authors to consider the reviewers' comments to improve the paper and resubmit to a future venue.